# Adaptation to constant light requires Fic-mediated AMPylation of BiP to protect against reversible photoreceptor degeneration

Andrew T Moehlman[1], Amanda K Casey[2,3,4], Kelly Servage[2,3,4], Kim Orth[2,3,4]*, Helmut Krämer[1,5]*

[1]Department of Neuroscience, UT Southwestern Medical Center, Dallas, United States; [2]Department of Molecular Biology, UT Southwestern Medical Center, Dallas, United States; [3]Department of Biochemistry, UT Southwestern Medical Center, Dallas, United States; [4]Howard Hughes Medical Institute, Dallas, United States; [5]Department of Cell Biology, UT Southwestern Medical Center, Dallas, United States

**Abstract** In response to environmental, developmental, and pathological stressors, cells engage homeostatic pathways to maintain their function. Among these pathways, the Unfolded Protein Response protects cells from the accumulation of misfolded proteins in the ER. Depending on ER stress levels, the ER-resident Fic protein catalyzes AMPylation or de-AMPylation of BiP, the major ER chaperone and regulator of the Unfolded Protein Response. This work elucidates the importance of the reversible AMPylation of BiP in maintaining the *Drosophila* visual system in response to stress. After 72 hr of constant light, photoreceptors of *fic*-null and AMPylation-resistant *BiP*[T366A] mutants, but not wild-type flies, display loss of synaptic function, disintegration of rhabdomeres, and excessive activation of ER stress reporters. Strikingly, this phenotype is reversible: photoreceptors regain their structure and function within 72 hr once returned to a standard light:dark cycle. These findings show that Fic-mediated AMPylation of BiP is required for neurons to adapt to transient stress demands.

DOI: https://doi.org/10.7554/eLife.38752.001

*For correspondence:
kim.orth@utsouthwestern.edu (KO);
helmut.kramer@utsouthwestern.edu (HK)

## Introduction

Post-translational modifications (PTMs) of proteins are important for rapid responses to environmental challenges of cells. One such PTM is AMPylation, the reversible addition of adenosine monophosphate (AMP) to hydroxyl groups (also known as adenylylation) (*Kingdon et al., 1967*; *Brown et al., 1971*; *Woolery et al., 2014*; *Casey and Orth, 2018*). AMPylation is catalyzed by at least two protein families, among them the conserved Fic-domain proteins (*Harms et al., 2016*; *Casey and Orth, 2018*). Eukaryotic Fic, an ER-resident type-II membrane protein (*Rahman et al., 2012*), AMPylates BiP (GRP78), a highly conserved and ubiquitous ER chaperone (*Ham et al., 2014*; *Preissler et al., 2015*). Working together with a multitude of associated quality control proteins, BiP is critical for the translocation, folding, and secretion of proteins from the ER as well as for aiding in the clearing of misfolded ER aggregates and degradation of membrane-associated proteins (*Hendershot et al., 1988*; *Kozutsumi et al., 1988*; *Meunier et al., 2002*). BiP is both a mediator and transcriptional target of the Unfolded Protein Response (UPR), a coordinated cell signaling pathway that is activated during times of high misfolded protein levels in the ER. Like many protein chaperones, BiP depends on its ATPase activity to undergo a conformational change to bind to its substrates (*Gaut and*

*Hendershot, 1993*). AMPylation locks BiP into a state resembling the ATP-bound conformation with high substrate off-rates, thereby inhibiting its chaperone function (*Preissler et al., 2017b*; *Wieteska et al., 2017*).

In agreement with this PTM's inhibitory role, BiP AMPylation levels are linked to protein homeostasis (*Ham et al., 2014*). Reduction of ER protein load promotes Fic-mediated AMPylation of BiP, whereas Fic catalyzes the deAMPylation of BiP under elevated ER stress conditions (*Ham et al., 2014*; *Preissler et al., 2015*). This switch in Fic's activity is linked to a key regulatory salt bridge in eukaryotic Fic. Mutations in Fic that disrupt this salt bridge result in an overactive AMPylator that lacks deAMPylation activity (*Casey et al., 2017*; *Preissler et al., 2017a*). Together, these studies suggest a model in which BiP is AMPylated in times of low ER stress, creating a reserve pool of inactive BiP that can be readily activated to respond to changes of ER homeostasis (*Figure 1A*). This reserve pool of BiP is proposed to act as a buffer to attenuate or shorten the need for a more dramatic activation of the transcriptional and translational arms of the UPR (*Casey et al., 2017*; *Preissler et al., 2017a*; *Wieteska et al., 2017*). However, the physiological importance of endogenous Fic-mediated AMPylation remains unclear.

In the fruit fly, *Drosophila melanogaster*, we previously demonstrated that *fic*-null mutants harbor a defect in visual signaling, as assessed by electroretinogram (ERG). The well-characterized Drosophila visual system has proven a valuable model for many fields, such as neuroscience (*Borycz et al., 2002*; *Sugie et al., 2015*), cell signaling (*Dolph et al., 1993*; *Scott et al., 1995*), protein trafficking (*Lee et al., 2003*; *Akbar et al., 2009*), and neurodegeneration (*Leonard et al., 1992*; *Johnson et al., 2002*; *Ryoo et al., 2007*). The specialized photoreceptor cells possess tightly packed microvilli-like membranes, termed rhabdomeres, that endow remarkable sensitivity to minute changes in light conditions (*Montell, 2012*). The ability to maintain this sensitivity is critical for flight behavior, foraging, and escape from predators. Thus, under a wide range of conditions, photoreceptors must maintain their light detection cascade, which requires the constant production, trafficking, and degradation of proteins through the endomembrane system (*Colley et al., 1995*; *Kiselev et al., 2000*; *Rosenbaum et al., 2006*).

We postulated that as a regulator of proteostasis and the UPR, BiP must be tightly regulated for proper maintenance of vision in the fly. Here we demonstrate that mutants lacking AMPylation of BiP at a specific residue, Thr366, possess the same ERG defects as *fic*-null animals, implicating BiP as the target of Fic necessary for visual signaling. We go on to find that application of an eye-specific stress, constant light, leads to exaggerated electrophysiology defects and the loss of photoreceptor rhabdomeres, akin to neurodegeneration. However, these defects exhibit a remarkable and unique reversibility: structural and functional phenotypes observed in light-stressed *fic*-null and AMPylation-resistant *BiP*T366A mutants are reversed after the flies are returned to a standard light/dark cycle. In addition, we identify changes in the regulation of UPR during constant light stress in these mutants, implicating dysregulation of ER homeostasis as a probable cause of the inability to adapt to altered light conditions.

## Results

### *BiP*T366A rescues over-expression of constitutively active AMPylating Fic^E247G

To test the hypothesis that BiP is a critical target of Fic AMPylation in vivo (*Figure 1A*), we designed and generated transgenes expressing wild-type and AMPylation-resistant FLAG-tagged BiP proteins under control of the endogenous BiP promoter (*Figure 1—figure supplement 1A*). *BiP* null fly mutants die early in development; this lethality is rescued by including a copy of the genomic transgene expressing FLAG-BiP^WT or the AMPylation-resistant FLAG-BiP^T366A mutant (*Figure 1B*). We will refer to these rescued flies as *BiP*^WT or *BiP*^T366A, respectively. In contrast, a BiP transgene mutated at a second reported AMPylation site (*Preissler et al., 2015*; *Casey et al., 2017*), BiP^T518A, did not rescue *BiP*^-/- lethality. As *fic* null mutants that lack BiP AMPylation are viable, the lethality of the *BiP*^T518A mutant is not likely to be due to the loss of AMPylation (*Casey et al., 2017*). Instead, these observations indicate an essential role for Thr^518 in BiP's chaperone activity. Consistent with this notion, the equivalent residue, Thr^538, in the *S. cerevisiae* BiP homolog Kar2 is required for survival

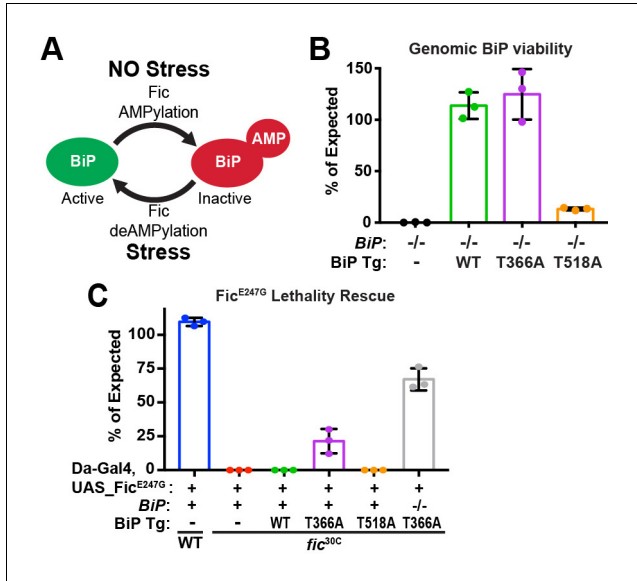

**Figure 1.** BiP is a target of Fic AMPylation and deAMPylation in vivo. (**A**) BiP AMPylation during times of low ER stress reserves a portion of the chaperone to allow for a rapid, deAMPylation-driven, response to high ER stress (**Casey et al., 2017**; **Preissler et al., 2017a**). (**B**) Bar graphs show the percentage of null mutant $BiP^{G0102}/y$ males rescued by the indicated genomic $BiP^{WT}$, $BiP^{T366A}$ or $BiP^{T518A}$ genomic transgene (Tg) relative to sibling controls. N = 3 biological replicas. At least 50 flies scored for each replica. Bar graphs show means ± Standard Deviation (SD). (**C**) Bar graphs show the percentage of viable flies of the indicated wild-type or $fic^{30C}$ genotypes expressing the overactive $Fic^{E247G}$ under the ubiquitous Da-Gal4 driver relative to sibling controls. Among the indicated genomic BiP transgenes, only $BiP^{T366A}$ provides partial rescue of lethality in the $BiP^{+/+}$ background and near complete rescue in a $BiP^{G0102}$ null background. N = 3 biological replicas. At least 100 total flies scored for each replica. Bar graphs show means ± SD.

DOI: https://doi.org/10.7554/eLife.38752.002

The following source data and figure supplements are available for figure 1:

**Source data 1.** Relates to *Figure 1B and C*.
DOI: https://doi.org/10.7554/eLife.38752.006
**Figure supplement 1.** Expression of genomic BiP transgenes.
DOI: https://doi.org/10.7554/eLife.38752.003
**Figure supplement 2.** AMPylation-resistant $BiP^{T366A}$ phenocopies *fic*.
DOI: https://doi.org/10.7554/eLife.38752.004
**Figure supplement 2—source data 1.** Relates to *Figure 1—figure supplement 2B*.
DOI: https://doi.org/10.7554/eLife.38752.007
**Figure supplement 3.** Red eye pigment suppresses ERG phenotypes of $fic^{30C}$ and $BiP^{T366A}$ mutants.
DOI: https://doi.org/10.7554/eLife.38752.005
**Figure supplement 3—source data 2.** Relates to *Figure 1—figure supplement 3B*.
DOI: https://doi.org/10.7554/eLife.38752.008

under heat stress even though yeast lack both Fic domain proteins and BiP AMPylation (*Figure 1—figure supplement 1B*).

Previously, we reported that over-expression of the constitutively active AMPylating $Fic^{E247G}$ was lethal in a *fic*-null fly background ($fic^{30C}$) because it lacks the essential deAMPylation activity (*Casey et al., 2017*). We tested whether flies expressing the AMPylation-resistant $BiP^{T366A}$ could survive this lethality. Consistent with previous findings, we observe over-expression of the mutant UAS-$Fic^{E247G}$ transgene using the ubiquitous *Da*-Gal4 driver was lethal in an otherwise *fic*-null animal (*Figure 1C*). Lethality of the constitutively active AMPylating $Fic^{E247G}$ was suppressed in flies expressing the AMPylation-resistant $BiP^{T366A}$ but not $BiP^{WT}$ (*Figure 1C*). These results indicate that $Thr^{366}$ of BiP is a physiologically relevant AMPylation target in vivo.

## The UPR protects eyes from overactive AMPylation

To test the interaction between Fic-mediated AMPylation and the UPR, we employed an eye-specific Fic gain-of-function model. Eye-specific expression of the constitutively active AMPylating UAS-Fic$^{E247G}$ transgene using a LongGMR-Gal4 driver in otherwise *fic*-null animals results in a severe rough-eye defect (*Casey et al., 2017*). However, in a *fic* heterozygous background, eye-specific expression of constitutively active AMPylating Fic$^{E247G}$ yields only a mildly rough eye (*Figure 2A*). We used this intermediate phenotype to assess genetic interactions between Fic$^{E247G}$ and components of the UPR with a candidate-based targeted RNAi screen (*Figure 2—figure supplement 1*). Fic$^{E247G}$-induced eye roughness was significantly enhanced by knockdown of the key UPR components *Perk*, *Atf4*, and *Ire1* (*Figure 2C–E* and *Figure 2—figure supplement 1*), but not *ATF6* (*Figure 2B*). Furthermore, *Xbp1* knockdown in conjunction with Fic$^{E247G}$ expression was lethal (*Figure 2F*). By contrast, knockdown of these UPR genes in the absence of Fic$^{E247G}$ did not influence eye phenotype or fly survival (*Figure 2A'–F'*). These genetic interactions suggest a role for UPR signaling in mitigating cellular stress imposed by expressing the constitutively active AMPylating Fic$^{E247G}$ in the eye.

## AMPylation of BiP is necessary for maintaining vision

The findings that BiP is a target of Fic in vivo and that silencing UPR pathway components enhances the severity of the constitutively active AMPylating Fic$^{E247G}$-induced rough-eye phenotype prompted us to assay the physiological effects of cellular stress in flies lacking BiP AMPylation. To do this we utilized flies that are either null for *fic* (*fic$^{30C}$*) or express the AMPylation-resistant BiP$^{T366A}$ instead of wild-type BiP. By using this strategy, we are able to discern BiP AMPylation-specific changes from other potential changes that are due to as-yet-unknown targets of Fic AMPylation.

As previously reported in ERG recordings, *fic*-null flies display a reduction of the initial response (termed the ON Transient, *Figure 3A*) to a light pulse compared to wild-type controls. Interestingly, BiP$^{T366A}$, but not BiP$^{WT}$ flies, exhibited the same defect in ON Transients as *fic$^{30C}$* mutants, consistent with BiP being the primary target of Fic AMPylation required for proper visual neurotransmission (*Figure 1—figure supplement 2*). Of note, we used an eye-specific RNAi construct against *white* to minimize any effect of the *mini-white* gene used as a marker in these BiP transgenes. When we compared ERG traces of *fic$^{30C}$* and BiP$^{T366A}$ flies in *white+* (red eyed) backgrounds, the

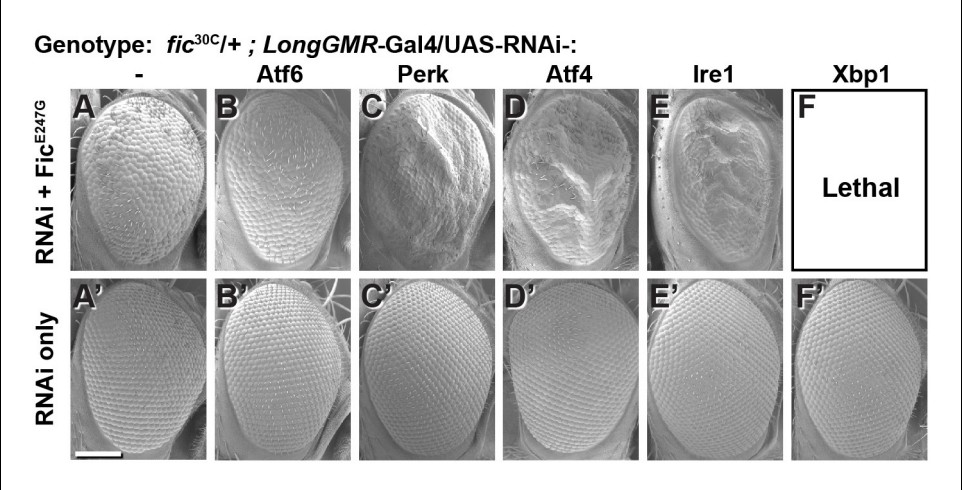

**Figure 2.** Genetic interactions between Fic and UPR genes. Representative SEM images of heterozygote mutant *fic$^{30C/+}$* eyes expressing the indicated UAS-RNAi transgenes with (**A–F**) or without (**A'–F'**) UAS-Fic$^{E247G}$ under *longGMR*-Gal4 control. See *Figure 2—figure supplement 1* for quantification. Scale bar: 100 μM.
DOI: https://doi.org/10.7554/eLife.38752.009
The following figure supplement is available for figure 2:

**Figure supplement 1.** Genetic interactions between Fic and UPR genes.
DOI: https://doi.org/10.7554/eLife.38752.010

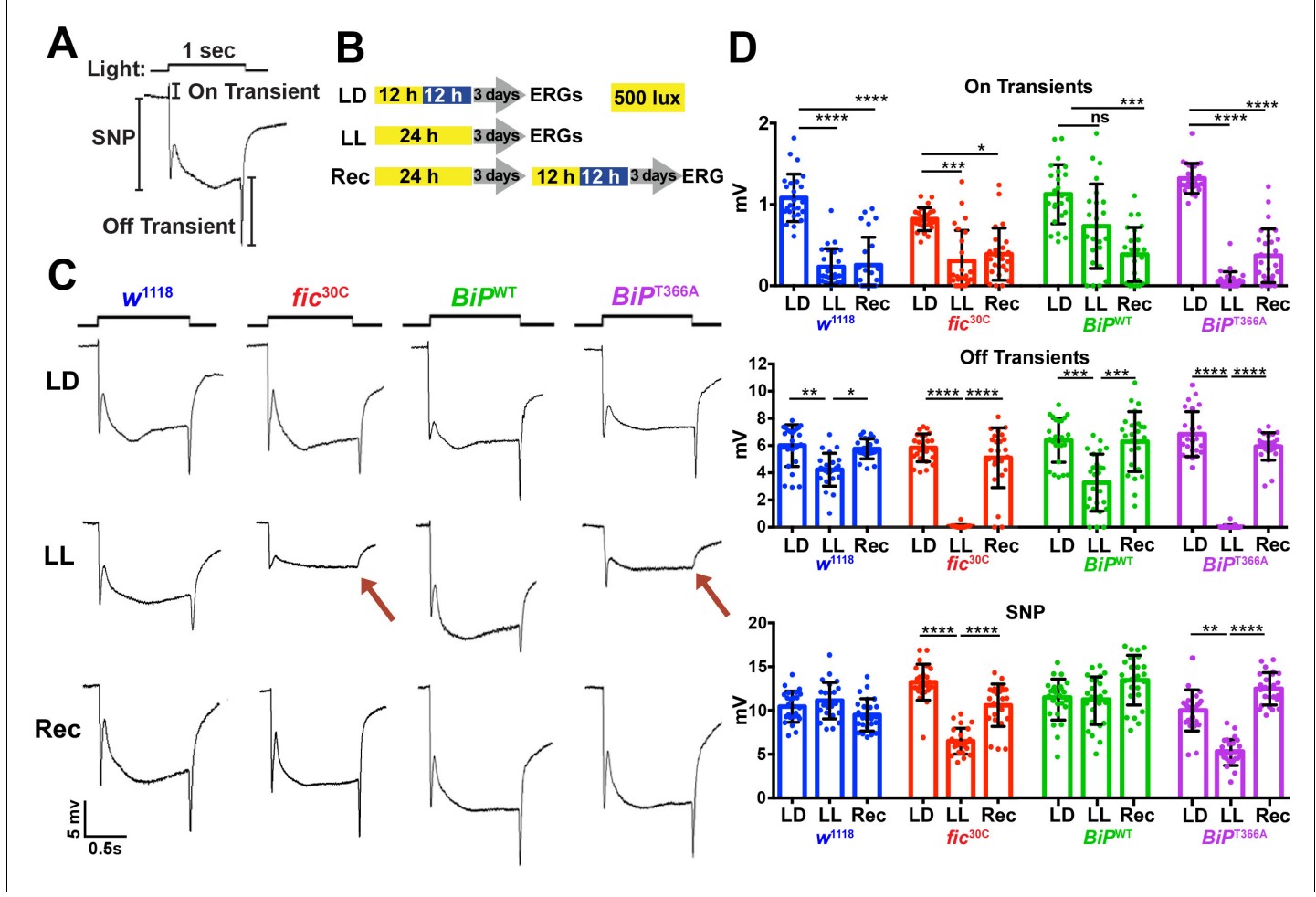

**Figure 3.** Fic-mediated AMPylation of BiP is required for photoreceptor maintenance. (**A**) A representative ERG trace in response to a 1 s light pulse displaying the sustained negative potential (SNP), representing the depolarization within photoreceptor neurons, and the ON and OFF transients, reflecting post-synaptic activity of lamina neurons. (**B**) Representation of the different light treatments of flies before ERG recordings: 3 days of 12 hr light (500 lux) and 12 hr dark (LD), 3 days of continuous light (LL) or 3 days of continuous light followed by 3 days of LD (Rec). 1 s light pulses were performed at 4 s intervals. (**C**) Representative traces from $w^{1118}$, $fic^{30C}$, $BiP^{WT}$ and $BiP^{T366A}$ flies. Under LL, $fic^{30C}$ and $BiP^{T366A}$ mutants lose ON and OFF transients (red arrows) and have reduced SNPs. The changes are reversed after 3 days of recovery (Rec). (**D**) Quantification of key components of ERGs shown in panel C. Bar graphs show means ± SD. ****p<0.0001; ***p<0.001; **p<0.01; *p<0.05; n = 24 flies for each genotype/condition, pooled from three independent biological replicas.

DOI: https://doi.org/10.7554/eLife.38752.011

The following source data and figure supplements are available for figure 3:

**Source data 1.** Relates to *Figure 3D*.
DOI: https://doi.org/10.7554/eLife.38752.015

**Figure supplement 1.** Light-induced defects in light-startle activity in $fic^{30C}$ mutants.
DOI: https://doi.org/10.7554/eLife.38752.012

**Figure supplement 1—source data 1.** Relates to *Figure 3—figure supplement 1A, B, C and D*.
DOI: https://doi.org/10.7554/eLife.38752.016

**Figure supplement 2.** $fic^{30C}$ mutants are sensitive to constant light, regardless of total intensity.
DOI: https://doi.org/10.7554/eLife.38752.013

**Figure supplement 2—source data 2.** Relates to *Figure 3—figure supplement 2B*.
DOI: https://doi.org/10.7554/eLife.38752.017

**Figure supplement 3.** $Fic^{30C}$ mutants recover ERG properties in 72 hours after 10 days of LL.
DOI: https://doi.org/10.7554/eLife.38752.014

**Figure supplement 3—source data 3.** Relates to *Figure 3—figure supplement 3B*.
DOI: https://doi.org/10.7554/eLife.38752.018

reductions in ON transients were no longer detectable (*Figure 1—figure supplement 3*). This is likely due to the previously established protective effect provided by the red pigment in *white*[+] flies. Indeed, a similar *white*-dependent phenotype has been reported for photoreceptor synaptic plasticity after prolonged light exposure (*Sugie et al., 2015*; *Damulewicz et al., 2017*), consistent with previous observations that a functional *white* gene masks some degenerative phenotypes in the retina (*Lee and Montell, 2004*; *Soukup et al., 2013*). Therefore, we used *white*-eyed flies to test whether AMPylation may play a role in this type of photoreceptor plasticity, which is induced by rearing flies in uninterrupted low light for 72 hr (*Sugie et al., 2015*; *Damulewicz et al., 2017*).

We conducted ERG recordings under different light conditions with four fly lines, $w^{1118}$, $fic^{30C}$, $BiP^{WT}$ and $BiP^{T366A}$ (*Figure 3B*). Compared to age-matched siblings reared under the standard 12 hr Light:12 hr Dark (LD) treatment, $fic^{30C}$ and $BiP^{T366A}$ flies reared for three days under continuous light (LL) at 500 lux exhibited severe ERG defects. This included reduction in the sustained negative potential (SNP), a measure of photoreceptor activation, and loss of ON and OFF transients, which reflect synaptic transmission to downstream L1/L2 lamina neurons (*Figure 3C and D*). Wild-type controls maintained healthy OFF transients following LL, but ON transients were reduced, reflecting the sensitivity of this component to various light conditions (*Figure 3D*). To test for behavioral consequences, we assayed $w^{1118}$ and $fic^{30C}$ flies after 72 hr of LD or LL treatment for light-induced startle behavior using single-fly activity chambers (*Ni et al., 2017*). Following a 5 min light pulse, LD-reared $fic^{30C}$ flies exhibited a startle response indistinguishable from control $w^{1118}$ flies, whereas $fic^{30C}$ flies, but not $w^{1118}$ flies, failed to respond to the light pulse after 72 hr of LL (*Figure 3—figure supplement 1*). Thus, Fic-mediated AMPylation is required to maintain vision acuity under LL conditions.

We next designed experiments to test whether these functional ERG changes in flies lacking AMPylation reflected light-induced neurodegeneration or a failure to adapt to constant stimulation. First, we asked if the LL-induced ERG defects of $BiP^{T366A}$ and $fic^{30C}$ flies were reversible. We reared mutant and control flies for 72 hr in LL followed by 72 hr of recovery in LD (referred to as 'Rec'; *Figure 3B*). This recovery period was sufficient to restore both healthy OFF transients and SNPs in $BiP^{T366A}$ and $fic^{30C}$ flies (*Figure 3C and D*). Second, we asked if the intensity of the light would exaggerate the defects of $BiP^{T366A}$ and $fic^{30C}$ flies. Exposure of mutant or control flies with 5000 lux, instead of 500 lux, did not alter the severity of ERG defects, indicating the changes were not simply a reflection of the increased amount of total light exposure during LL treatment (*Figure 3—figure supplement 2*). Third, we asked if prolonging the LL stress would alter the reversibility of these defects. Mutant flies reared under LL for ten days retained the capability to recover healthy ERG traces after only three days on LD, indicating that photoreceptors are not dying but maintained during prolonged light stress (*Figure 3—figure supplement 3*). Together, these data support a model for a dysregulated adaptive response, rather than phototoxicity, inducing the reversible loss of OFF transients and reduced SNPs.

## Constant light induces severe but reversible morphological defects in AMPylation mutants

To determine if the underlying eye substructures were being altered in these AMPylation deficient mutants, we performed TEM on ultrathin transverse eye sections. Under LD conditions, $fic^{30C}$ and $BiP^{T366A}$ mutant and wild-type controls appeared indistinguishable (*Figure 4A*). However, following 72 hr of LL (500 lux), $fic^{30C}$ and $BiP^{T366A}$ mutants, but not $w^{1118}$ and $BiP^{WT}$ controls, displayed severe defects in the integrity of rhabdomeres, the microvilli-like membrane structures that house the phototransduction cascade (*Figure 4B*). After a three-day recovery at LD, the rhabdomeres were nearly restored in both AMPylation-deficient mutants (*Figure 4C*). To quantify these structural changes in large cohorts of flies, we assessed flies for the presence of wild-type 'deep pseudopupils' (DPP) (*Figure 4D*). Visualization of the DPP affords an assessment of rhabdomere structural integrity in living flies (*Franceschini and Kirschfeld, 1971*). Consistent with the TEM data, 3 days of LL caused loss of DPP in $fic^{30C}$ and $BiP^{T366A}$, and DPPs returned after a 3 day recovery (*Figure 4D*). This suggests that proper regulation of BiP through AMPylation is required for maintaining both function and structure of photoreceptor cells.

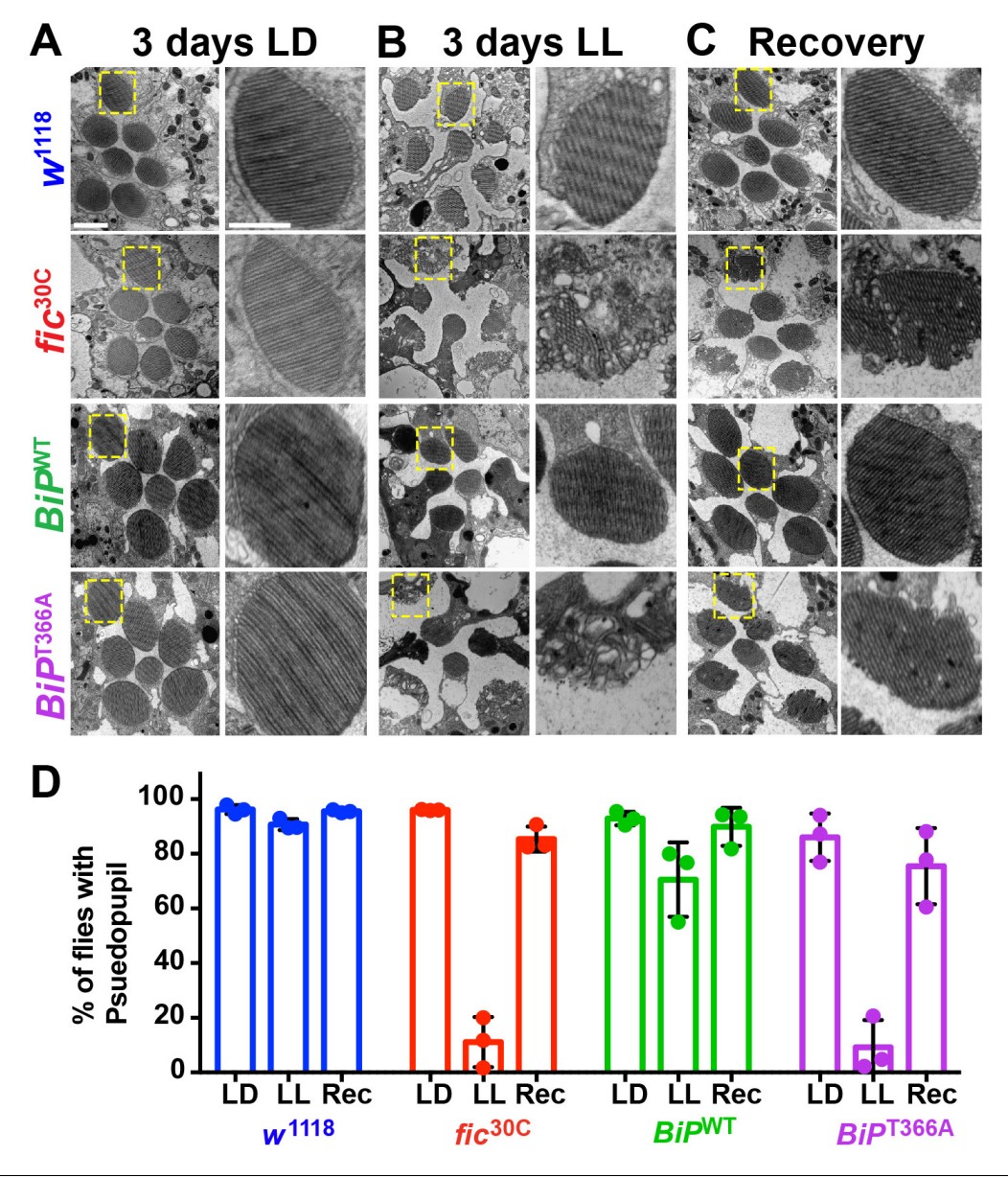

**Figure 4.** AMPylation of BiP is required for maintaining structural plasticity in the retina. (**A–C**) Representative TEM images of retina thin sections from the indicated genotypes with either standard LD (**A**), the stress-inducing LL (**B**) or recovery treatment (**C**). Scale bars: 1 µM. Yellow boxes indicate rhabdomeres shown in high magnification images. High magnification scale bars: 0.5 µM. (**D**) Percentages of flies with intact deep pseudopupil following LD, LL and Rec. N = 3 independent biological replicas with approximately 50 flies scored per genotype per replica. Bar graphs show means ± SD.

DOI: https://doi.org/10.7554/eLife.38752.019

The following source data is available for figure 4:

**Source data 1.** Relates to *Figure 4D*.

DOI: https://doi.org/10.7554/eLife.38752.020

# Fic is required for ER homeostasis in the visual system during constant light

Given the unique role of Fic in both AMPylating and deAMPylating BiP to modulate its chaperone activity and maintaining ER homeostasis, we evaluated $fic^{30C}$ flies for changes in the UPR under LD, LL, and Rec conditions. First, we performed immunohistochemistry against BiP, a transcriptional target of the UPR, which is upregulated during states of ER stress (*Gardner et al., 2013*; *Ham et al., 2014*). After 3 days of LL, sections of $fic^{30C}$ revealed increased levels of BiP in retinas and in the lamina neuropils where photoreceptor axons form synapses onto lamina neurons. BiP levels returned to control levels following three days of recovery (*Figure 5A and B*). To further assess UPR signaling in these tissues, we utilized a sensor for Ire1-mediated Xbp1 splicing (*Sone et al., 2013*) and an Atf4 translational reporter which serves as a proxy for Perk-mediated phosphorylation of eIF-2a (*Kang et al., 2015*). In wild-type flies, Xbp1-GFP was slightly induced in the lamina after 24 hr of LL in wild-type flies and the signal decreased over time (*Figure 5C*, top row, and *Figure 5D*). However, in $fic^{30C}$ flies, the Xbp1-GFP signal in the lamina continued to increase after 48 hr of LL and remained elevated after 72 hr (Figure bottom row, and *Figure 5D*). In the retina, control flies showed little to no increase of Xbp1-GFP levels, while $fic^{30C}$ flies showed a significant transient increase after one and two days LL. With the Atf4-DsRed reporter, we observed a significant increase of signal in both the retina and lamina of wild-type flies after one day, but no difference in $fic^{30C}$ mutants at one or two days LL when compared to LD controls (*Figure 5E and F*). However, by three days of LL, Atf4-DsRed reporter activity in the wild-type flies returned to control levels, while the $fic^{30C}$ mutants showed a significant increase in both the retina and lamina neuropil (*Figure 5E and F*). These changes in UPR signaling were reversible as each of the reporters returned to near control intensity after 72 hr of LD recovery (*Figure 5A,C and E*, last columns). The elevated UPR response in $fic^{30C}$ mutants correlated with the timing of the observed severe defects in the integrity of rhabdomeres (*Figure 4B*). Together, these data identify a role for Fic-mediated BiP AMPylation in regulating ER stress during homeostatic responses of the visual system.

## Discussion

Here we demonstrate that BiP is a critical in-vivo target of Fic-mediated AMPylation, as an AMPylation-resistant BiP blocks lethality caused by over-expressed constitutively active AMPylating $Fic^{E247G}$ and recapitulates *fic*-null vision defects. This work also sheds light on the physiological connection between AMPylation/deAMPylation of BiP and the UPR. We observe genetic interactions with the constitutively active AMPylating $Fic^{E247G}$ and the UPR sensors Ire1 and Perk as well as their effectors, perhaps due to the critical role of BiP as both a regulator (*Bertolotti et al., 2000*; *Shen et al., 2005*; *Carrara et al., 2015*; *Amin-Wetzel et al., 2017*) and downstream transcriptional target of the UPR (*Kozutsumi et al., 1988*; *Ham et al., 2014*). Indeed, we hypothesize that unregulated $Fic^{E247G}$, in the absence of deAMPylation activity, AMPylates excess BiP, excluding it from its normal chaperone role and leading to cell death (*Casey et al., 2017*; *Truttmann et al., 2017*). In support of this hypothesis, the developmental defects due to excessive unregulated AMPylation are suppressed by AMPylation-resistant $BiP^{T366A}$. Furthermore, the enhancement of the rough-eye $Fic^{E247G}$ phenotype by knockdown of the Ire1 and PERK pathways suggest a protective role for the UPR, perhaps through the known effects on translation by Ire1-mediated decay of mRNA, Xbp1-driven transcription or Perk-mediated phosphorylation of eIF-2a (*Gardner et al., 2013*).

Our work supports an in-vivo requirement for Fic-mediated AMPylation of $BiP^{T366}$ in the context of long-term adaptation to prolonged light exposure. $BiP^{T366A}$ replacement mutants phenocopy *fic*-null flies in both the light-induced blindness and the unexpected recovery from this phenotype. We observe changes in the ON/OFF transients reflecting defects in transmission to the downstream L1/L2 neurons and also a decreased magnitude in the SNP of photoreceptors. Functional changes in the SNP component are mirrored in the structural changes of photoreceptor rhabdomeres. Rhabdomere appearance of AMPylation-deficient flies was reminiscent of retinal degeneration mutants (*Smith et al., 1991*; *Ryoo et al., 2007*), however the remarkable recovery of the tissue structure in three days of LD is unlike any reported retinal degeneration phenotype. It is worth emphasizing that both rhabdomere structure and synaptic transmission recover, as these are not necessarily linked. For example, some mutant alleles of *ninaE*, encoding the R1-6 rhodopsin, or *rdgC*, encoding a rhodopsin phosphatase, maintain synaptic transmission despite substantial rhabdomere degeneration

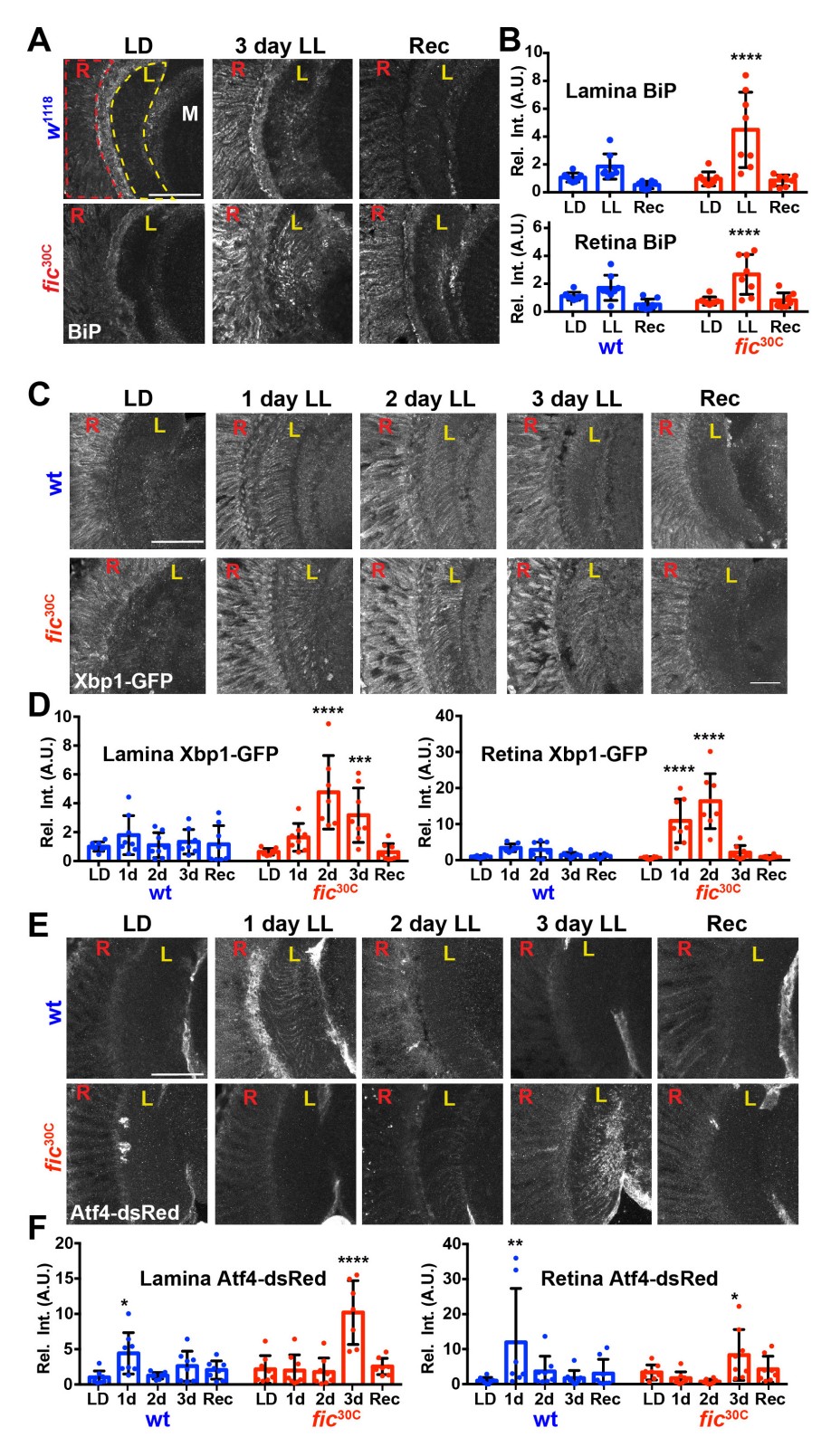

**Figure 5.** ER homeostasis is disturbed in *fic* mutants during prolonged light stimulation. (**A**) Representative images of BiP immunohistochemistry in sections of $w^{1118}$ and $fic^{30C}$ flies following 3 days LD, LL or Recovery treatments. (**B**) Quantification of BiP fluorescence intensity, normalized to wild-type LD controls, in the lamina neuropil and retina from two independent experiments. (**C**) Representative images of a Xbp1-GFP splicing reporter in either a Fic wild-type or the null $fic^{30C}$ background following LD, 1 day LL, 2 day LL, 3 day LL, and Recovery conditions. (**D**) Quantification of GFP fluorescence

*Figure 5 continued on next page*

*Figure 5 continued*

intensity, normalized to wild-type LD controls, in the lamina neuropil and retina from two independent experiments. (E) Representative images of an *Atf4*-dsRed reporter in either a wild-type or *fic*[30C] background following LD, 1 day LL, 2 day LL, 3 day LL, and Recovery conditions. (F) Quantification of *Atf4*-dsRed intensity, normalized to wild-type LD controls, in the lamina neuropil and retina from two independent experiments. For all experiments, n = 8 flies per genotype/condition, with exceptions of outliers falling three standard deviations outside the mean. Bar graphs show means ± SD. For all experiments, significance is indicated for treatment compared to the LD condition for the corresponding genotype. ****p<0.0001; ***p<0.001; **p<0.01; *p<0.05. All scale bars: 50 μM.

DOI: https://doi.org/10.7554/eLife.38752.021
The following source data is available for figure 5:

**Source data 1.** Relates to *Figures 5B, D and F*.
DOI: https://doi.org/10.7554/eLife.38752.022

(*Webel et al., 2000*; *Lee and Montell, 2001*). Together, our findings demonstrate a seminal role for Fic-mediated AMPylation of BiP in vivo: enabling photoreceptors to adapt and maintain both structural and functional integrity during periods of prolonged stress due to uninterrupted light stimulation. The exact mechanism through which these defects in *fic* mutants arise remains undetermined, but previous studies have demonstrated a requirement for maintaining normal ER folding and trafficking of transmembrane visual signaling components such as Rhodopsin (*Colley et al., 1995*; *Rosenbaum et al., 2006*). We hypothesize this intense demand for proper ER stress regulation sensitizes the eye to otherwise mild defects in *fic* mutants, and the additional demands posed by the stress of constant light stimulation.

We also observed that loss of BiP AMPylation deregulates, but does not block, the UPR to this physiological stress. This observation supports previous claims that AMPylation and deAMPylation of BiP aids in maintaining ER homeostasis (*Figure 1A*) by establishing a reserve pool of BiP that can readily be deAMPylated in response to acute ER insults (*Ham et al., 2014*; *Casey et al., 2017*; *Preissler et al., 2017a*). This regulation would allow for fine-tuning of the UPR response under specific contexts, comparable to findings in *C. elegans* in which *fic-1* and *hsp3* (a BiP homologue) mutants are sensitive to bacterial infection (*Truttmann et al., 2016*). We speculate that the eye requires tight control of BiP activity, to facilitate adaptation of the vision signaling cascade. Under standard LD conditions, only slight differences are observed, presumably because ER stress is low (*Figure 5C and D*, first column). However, exposure to constant light results in ER stress and UPR signaling which wild-type flies can clear, presumably because there is a reserve pool of AMPylated BiP to rapidly respond to the stress. In *fic* mutants, we speculate, loss of the reserve BiP results in the UPR dysregulation revealed by the Ire1 and Perk activity reporters (*Figure 5C and D*) as a larger proportion of BiP would be previously engaged and not able to respond to the extra stress. Additionally, the prolonged UPR response in the eyes with dysregulated AMPylation may result in increased expression of UPR-regulated proteins while blocking production of the visual signaling components necessary for adapting to transient stress. It is unknown whether the ratio of AMPylated to unmodified BiP directly regulates activation of the UPR sensors, given the multiple levels of feedback on the system (*Gardner et al., 2013*). It remains to be determined whether the effects on UPR signaling are the direct cause of the visual system defects or if the phenotypes in *fic* and *BiP*[T366A] mutants are primarily due to changes in protein folding and secretion. We also cannot rule out the influence of AMPylation on other ER processes that involve BiP, such as ER-associated degradation (*Hegde et al., 2006*).

BiP expression is subject to multiple levels of feedback regulation and is induced when the UPR is activated (*Kozutsumi et al., 1988*; *Ma and Hendershot, 2003*). However, in a negative feedback loop, BiP also inhibits activation of the UPR sensors Ire1, Perk, and Atf6, through direct binding (*Bertolotti et al., 2000*; *Shen et al., 2005*). It remains unknown how AMPylation affects the interactions of BiP with these proteins in vivo; however, in-vitro work suggests that AMPylation of BiP abolishes its inhibitory effect on Ire1 dimerization and activation (*Amin-Wetzel et al., 2017*). We speculate that the loss of BiP AMPylation/deAMPylation cycle in a *fic* null affects both the ability of BiP to quickly respond to misfolded protein aggregates and to regulate UPR activation. Future studies on the dynamic role of reversible BiP AMPylation and its interaction with downstream UPR sensors should provide unique insight into neuronal plasticity and regeneration.

# Materials and methods

**Key resources table**

| Reagent type (species) or resource | Designation | Source or reference | Identifiers | Additional information |
|---|---|---|---|---|
| Gene (*Drosophila melanogaster*) | *fic* | NA | FLYB:FBgn0263278 | |
| Gene (*D. melanogaster*) | *Hsc3-70* | NA | FLYB:FBgn0001218 | |
| Genetic reagent (*D. melanogaster*) | Da-Gal4 | Bloomington *Drosophila* Stock Center | BDSC:55851; FLYB:FBst0055851; RRID: BDSC_55851 | |
| Genetic reagent (*D. melanogaster*) | LongGMR-Gal4 | Bloomington *Drosophila* Stock Center | BDSC:8121; FLYB:FBst0008121; RRID:BDSC_8121 | |
| Genetic reagent (*D. melanogaster*) | W[1118] | Bloomington *Drosophila* Stock Center | BDSC:3605; FLYB:FBst0003605; RRID:BDSC_3605 | |
| Genetic reagent (*D. melanogaster*) | BiP[G0102]/FM7c | Bloomington *Drosophila* Stock Center | BDSC:11815; FLYB:FBal0098203; RRID:BDSC_11815 | |
| Genetic reagent (*D. melanogaster*) | UAS[Scer]-Xbp1-GFP.hg | Bloomington Drosophila Stock Center | BDSC:60731; FLYB:FBst0060731; RRID:BDSC_60731 | *Sone et al. (2013)* |
| Genetic reagent (*D. melanogaster*) | dsRed.crc(ATF4).5'UTR.tub | DOI: 10.1371/journal.pone.0126795; PMID:25978358 | FLYB: FBal0304834 | Gift from Don Ryoo, NYU. Tubulin promoter and ATF4 5'UTR drive DsRed expression (Flybase FBal0304834) |
| Genetic reagent (*D. melanogaster*) | genomic 3xFLAG-BiP[WT] | This paper | | pAttb_gen3xFLAG-BiP[WT] inserted in AttP landing site at 89E11 |
| Genetic reagent (*D. melanogaster*) | genomic 3xFLAG-BiP[T366A] | This paper | | pAttb_gen3xFLAG-BiP[T366A] inserted in AttP landing site at 89E11 |
| Genetic reagent (*D. melanogaster*) | genomic 3xFLAG-BiP[T518A] | This paper | | pAttb_gen3xFLAG-BiP[T518A] inserted in AttP landing site at 89E11 |
| Genetic reagent (*D. melanogaster*) | *GMR*-dsRNA[white] | This paper | | pAttb_GMR-dsRNA[white] inserted in AttP landing site at 43A1 |
| Genetic reagent (*D. melanogaster*) | *fic*[30C] | DOI: 10.1074/jbc.M117.799296; PMID:29089387 | | *Casey et al. (2017)* |
| Genetic reagent (*D. melanogaster*) | UAS[Scer]-V5-Fic[E247G] | DOI: 10.1074/jbc.M117.799296; PMID:29089387 | | *Casey et al. (2017)* |
| Genetic reagent (*D. melanogaster*) | | | | UPR and ER protein RNAi lines screened are contained in Supplemental Table 1 |
| Genetic reagent (*Saccharomyces cerevisiae*) | KAR2::KAN | GE Healthcare Life Sciences | SGD:S000003571 | gene replacement generated using PCR-based gene deletion strategy yielding start- to stop-codon deletion |
| Strain, strain background (*Saccharomyces cerevisiae*) | BY4741 MATa his3Δ1 leu2Δ0 met15Δ0 ura3Δ0 | DOI: 10.1002/(SICI)1097-0061(19980130)14:2 < 115::AID-YEA204 > 3.0.CO;2–2; PMID: 9483801 | GenBank: JRIS00000000.1 | |
| Antibody | anti-Hsc70-3 (BiP) (Guinea Pig polyclonal) | DOI: 10.1038/sj.emboj.7601477; PMID:17170705 | FLYB: FBgn0001218; RRID: AB_2569409 | Gift from Don Ryoo, NYU (1:2000 IHC, 1:8000 WB) |
| Antibody | anti-RFP (Rabbit polyclonal) | Rockland | Rockland:600-401-379; RRID: AB_2209751 | (1:1000 IHC) |

*Continued on next page*

*Continued*

| Reagent type (species) or resource | Designation | Source or reference | Identifiers | Additional information |
|---|---|---|---|---|
| Antibody | anti-GFP (Chicken polyclonal) | ThermoFisher Scientific | ThermoFisher Scientific:A10262; RRID: AB_2534023 | (1:1000 IHC) |
| Antibody | anti-Flag (mouse monoclonal) | Sigma | Sigma:F-3165; RRID:AB_259529 | (1:2000 WB) |
| Antibody | anti-Actin (mouse monoclonal) | Developmental Studies Hybridoma Bank | DSHB:JLA20; RRID: AB_528068 | 1:2000 (WB) |
| Antibody | Alexa 488- or 568- secondaries | Molecular Probes | | (1:1000 IHC) |
| Antibody | LICOR 800 or 700- secondaries | LICOR Biosciences | | (1:20,000 WB) |
| Recombinant DNA reagent | pAttb_gen3xFLAG-BiP[WT] | This paper | | PCR in multiple steps from genomic DNA {sequence location = X: 11,801,696..11,807,117 [-]}. Cloned into modified pAttb vector |
| Recombinant DNA reagent | pAttb_gen3xFLAG-BiP[T366A] | This paper | | Progenitors: pAttb_gen3xFLAG-BiP[WT]. Mutated sequence synthesized with Geneblock (IDT) |
| Recombinant DNA reagent | pAttb_gen3xFLAG-BiP[T518A] | This paper | | Progenitors: pAttb_gen3xFLAG-BiP[WT]. Mutated sequence synthesized with Geneblock (IDT) |
| Recombinant DNA reagent | pAttb_GMR-dsRNA[white] | This paper | | Progenitors: pUASt_dsRNA[white] (gift from Dean Smith, UT Southwestern, PMID: 11804566). GMR sequence: Geneblock (IDT) |
| Recombinant DNA reagent | pKAR2:LEU2 | This paper | | Cloned from amplification of endogenous KAR2 with primers (see below) |
| Recombinant DNA reagent | pKAR2[T386A]:LEU2 | This paper | | Progenitor: pKAR2:LEU2. Site directed mutagenesis used to make mutation |
| Recombinant DNA reagent | pKAR2[T538A]:LEU2 | This paper | | Progenitor: pKAR2:LEU2. Site directed mutagenesis used to make mutation |
| Sequence-based reagent | 5'-GCATCCGCGGATACTCTCGTACCCTGCCGC-3' | This paper | | Cloning for pKAR2:LEU2 |
| Sequence-based reagent | 5'-ATGCGAGCTCCGTATATACTCAGTATAATC-3' | This paper | | Cloning for pKAR2:LEU2 |
| Sequence-based reagent | 5'-GGTTGGTGGTTCTGCTAGAATTCCAAAGGTCCAACAATTGTTAGAATCATACTTTGATGG-3' | This paper | | Mutagenesis primer for pKAR2[T386A]:LEU2 |
| Sequence-based reagent | 5'-ACCTTTGGAATTCTAGCAGAACCACCAACCAAAACGATATCATCAACATCCTTCTTTTCC-3'. | This paper | | Mutagenesis primer for pKAR2[T386A]:LEU2 |
| Sequence-based reagent | 5'-AGATAAGGGAGCTGGTAAATCCGAATCTATCACCATCACTAACG-3' | This paper | | Mutagenesis primer for pKAR2[T538A]:LEU2 |
| Sequence-based reagent | 5'-GGATTTACCAGCTCCCTTATCTGTGGCAGACACCTTCAGAATACC-3'. | This paper | | Mutagenesis primer for pKAR2[T538A]:LEU2 |
| Chemical compound, drug | VECTASHIELD Antifade Mounting Medium with DAPI | Vector Laboratories | Vector Laboratories: H-1200 | |
| Software, algorithm | Adobe Photoshop | Adobe | RRID:SCR_014199 | |
| Software, algorithm | ImageJ | NIH | RRID:SCR_003070 | |

## Fly stocks and genetics

Bloomington Stock Center provided $w^{1118}$ (BS# 3605), $BiP^{G0102}$/FM7c (BS#11815), Da-Gal4 (BS#55850), LongGMR-Gal4 (BS#8121) stocks. The $fic^{30C}$ allele and UAS-Fic$^{E247G}$ flies was previously described (Casey et al., 2017). Lines used in the RNAi screen are described in Figure 2—figure supplement 1 and were obtained from Bloomington Stock Center and the Vienna Drosophila Resource Center (Dietzl et al., 2007). The Atf4$^{5'UTR}$-dsRed (Kang et al., 2015) and the Xbp1-GFP (Sone et al., 2013) lines were a gift from Dr. Don Ryoo (NYU) and were recombined with the $fic^{30C}$ allele. We generated the p[gen3xFLAG-BiP$^{WT}$]$^{AttP-89E11}$, p[gen3xFLAG-BiP$^{T366A}$]$^{AttP-89E11}$, p[gen3-xFLAG-BiP$^{T518A}$]$^{AttP-89E11}$ and p[GMR-dsRNA$^{white}$] alleles using the Phi30C integrase strategy (Venken et al., 2006). p[GMR-dsRNA$^{white}$] was recombined with the $fic^{30C}$ allele and white-eyed candidates were screened for the fic allele by PCR. $BiP^{G0102}$;; p[gen3xFLAG-BiP$^{WT}$]$^{AttP-89E11}$ and $BiP^{G0102}$;;p[gen3xFLAG-BiP$^{T366A}$]$^{AttP-89E11}$ stocks were made by crossing males harboring the genomic transgene to $BiP^{G0102}$/FM7c female flies. Surviving males were backcrossed to $BiP^{G0102}$/FM7c female flies, and stable stocks were established from the resulting progeny. None of the rare escaping $BiP^{G0102}$; ; p[gen3xFLAG-BiP$^{T518A}$]$^{AttP-89E11}$ male flies were fertile. The LongGMR-Gal4,UAS$^{Scer}$-V5-Fic$^{E247G}$$^{-attP-B3}$/TM6B,Hu and Da-Gal4,UAS$^{Scer}$-V5-Fic$^{E247G}$$^{-attP-3B}$/TM6B,hu stocks were made using standard Drosophila recombination and crossed into $w^{1118}$ and $w^{1118}$; $fic^{30C}$ backgrounds.

## List of fly strains and stocks

$w^{1118}$ (BS#3605)

OreR

; $fic^{30C}$

$w^{1118}$; $fic^{30C}$

$w^{1118}$; p[GMR-dsRNA$^{white}$]

$w^{1118}$; $fic^{30C}$, p[GMR-dsRNA$^{white}$]

$w^{1118}$; $fic^{30C}$; LongGMR-Gal4,UAS$^{Scer}$-V5-Fic$^{E247G-}$ $^{attP-3B}$/TM6B,hu

$w^{1118}$; $fic^{30C}$/CyO; Da-Gal4,UAS$^{Scer}$-V5-Fic$^{E247G-}$ $^{attP-3B}$/TM6B,hu

$BiP^{G0102}$/FM7c (BS#11815)

$w^{1118}$;; p[gen3xFLAG-BiP$^{WT}$]$^{AttP-89E11}$

$w^{1118}$;; p[gen3xFLAG-BiP$^{T366A}$]$^{AttP-89E11}$

$w^{1118}$;; p[gen3xFLAG-BiP$^{T518A}$]$^{AttP-89E11}$

$w^{1118}$; $fic^{30C}$; p[gen3xFLAG-BiP$^{WT}$]$^{AttP-89E11}$

$w^{1118}$; $fic^{30C}$; p[gen3xFLAG-BiP$^{T366A}$]$^{AttP-89E11}$

$w^{1118}$; $fic^{30C}$; p[gen3xFLAG-BiP$^{T518A}$]$^{AttP-89E11}$

$BiP^{G0102}$; ; p[gen3xFLAG-BiP$^{WT}$]$^{AttP-89E11}$

$BiP^{G0102}$; ; p[gen3xFLAG-BiP$^{T366A}$]$^{AttP-89E11}$

$BiP^{G0102}$; p[GMR-dsRNA$^{white}$]; p[gen3xFLAG-BiP$^{WT}$]$^{AttP-89E11}$

$BiP^{G0102}$; p[GMR-dsRNA$^{white}$]; p[gen3xFLAG-BiP$^{T366A}$]$^{AttP-89E11}$

$w^{1118}$; p[tub-Atf4$^{5'UTR}$-dsRed]/p[GMR-dsRNA$^{white}$]

$w^{1118}$; $fic^{30C}$,p[tub-Atf4$^{5'UTR}$-dsRed]/fic$^{30C}$,p[GMR-dsRNA$^{white}$]

$w^{1118}$; p[GMR-dsRNA$^{white}$]; p[Da-Gal4]/p[UAS$^{Scer}$-Xbp1-GFP.hg]

$w^{1118}$; $fic^{30C}$,p[GMR-dsRNA$^{white}$]; p[Da-Gal4]/p[UAS$^{Scer}$-Xbp1-GFP.hg]

## Generation of genomic BiP transgenes

PCR-amplified BiP genomic sequences were cloned into a pAttB vector and a 3X-FLAG tag was inserted after the N-terminal signal sequence. To create the T366A and T518A mutations, gBlocks (IDT, Coralville, IA) for the mutant sequences were synthesized and subcloned into the pAttB_genomic BiP vector via NEB HiFi Assembly Kit (NEB, Ipswich, MA). These constructs were sequence-verified and injected into embryos (BestGene, Chino Hills, CA) for insertion at the 89E11 landing site. Expression levels of FLAG-BiP transgenes were determined with western blotting. In brief, fly heads were homogenized in lysis buffer (10% SDS, 6M urea, and 50 mM Tris-HCl, pH 6.8 + 10% DTT), sonicated for 5 min, boiled for 2 min, and centrifuged for 10 min at 10,000 g to remove debris. 10 μL were separated by SDS-PAGE and transferred to nitrocellulose membranes. Blots were probed with anti-BiP (1:8000, gift from Dr. Don Ryoo, NYU, NY), anti-FLAG (1:2000 M2- F3165,

Millipore Sigma, St. Louis, MO) and anti-Actin (1:4000, JLA-20, DSHB, Iowa City, IA) and detected using IRdye-labeled antibodies and an Odyssey scanner (LI-COR Biosciences, Lincoln, NE).

## Generation of GMR_dsRNA[white] transgenes

To make the eye-targeted dsRNA constructs against the *white* gene, the dsRNA sequence was obtained from a pAttb-UAS[S.cer]-dsRNA[white] vector (*Kalidas and Smith, 2002*) (a gift from Dr. Dean Smith, UT Southwestern Medical Center, TX) and the UAS[S.cer]-Hsp40 promotor sequence was replaced with a 5X-*GMR* promotor sequence, synthesized as a gBlock (IDT) and cloned with NEB HiFi Assembly Kit (NEB).

## Fly rearing conditions

All flies were reared on standard molasses fly food, under room temperature conditions. For light treatments, flies were collected within one to two days of enclosing, and placed in 5 cm diameter vials containing normal food, with no more than 25 flies, and placed at either LD (lights ON 8am/ lights OFF 8pm) or LL. ERGs, head dissections and behavior assays were performed between 1pm and 4pm. The same intensity white LED light source was used for both conditions and flies were kept the same distance away from the light source, which amounted to approximately 500 lux. LD and LL treatments were done at 25°C. For the UPR and Fic[E247G] rough-eye interaction experiments, all flies were raised at 28°C.

## Survival analysis of flies expressing genomic BiP construct

*BiP*[G0102]/FM7c female virgin flies were crossed to males with either gen3xFLAG-BiP[WT], gen3xFLAG-BiP[T366A] or gen3xFLAG-BiP[T518A]. The number of surviving non-FM7c male flies was scored by presence or lack of the *Bar* eye marker. Percent of expected was calculated from the actual number or recovered flies of the relevant genotypes compared the expected Mendelian number [# observed flies/ #expected flies]. Crosses were repeated three times (n = 3). Total number of flies scored was at least 100 for each BiP variant.

## Survival analysis of flies expressing BiP variants in a *Da*-Gal4, UAS-Fic[E247G] background

C-terminally V5-His6-tagged UAS-Fic[E247G] (*Casey et al., 2017*) was expressed via the ubiquitous *Da*-Gal4 driver in *fic*[30C]/CyO heterozygous flies. These flies were crossed to w[-]; *fic*[30C] (controls), w[-]; *fic*[30C]; gen3xFLAGBiP[WT], w-; *fic*[30C]; gen3xFLAG[T366A], w-; *fic*[30C]; gen3xFLAG-BiP[T518A], or *BiP*[G0102]; *fic*[30C]; gen3xFLAG[T366A]. Offspring were scored and the number of adults homozygous for *fic*[30C] with the *Da*-Gal4, UAS-Fic[E247G] allele and were compared to the number of *fic*[30C] heterozygous sibling controls. Percent of expected was calculated from the actual number or recovered flies of the relevant genotypes compared the expected Mendelian number [# observed flies/ #expected flies]. Crosses were repeated three times (n = 3). Total number of flies scored was at least 100 for each BiP variant each repeat.

## Electroretinograms

ERGs were recorded as previously described (*Montell, 2012*). Glass electrodes filled with 2M NaCl were placed in the fly thorax and surface of the corneal lens (recording). A computer-controlled LED light source (MC1500; Schott, Mainz, Germany) was pulsed for 1 s at 4 s intervals. The resulting ERG traces were collected by an electrometer (IE-210; Warner Instruments, Hamden, CT), digitized with a Digidata 1440A and MiniDigi 1B system (Molecular Devices, San Jose, CA), and recorded using Clampex 10.2 (Molecular Devices) and quantified with Clampfit software (Molecular Devices). Flies were assayed in batches of eight to ten, and resulting quantifications are pooled from three independent biological repeats.

## Deep pseudopupil analysis

Flies were anesthetized on $CO_2$ and aligned with one eye facing up. Using a stereoscopic dissection microscope, each fly was scored for presence or loss of the deep pseudopupil (*Franceschini and Kirschfeld, 1971*), and the percentage of flies with intact pseudopupils was calculated. For each

genotype/treatment, over 50 flies were scored per replica and three biological replicas were performed (n = 3).

## Light-startle behavior assay

Assay was adapted from a previously described method (*Ni et al., 2017*). After 72 hr of LD or LL treatment, 16 flies per genotype were collected at the same time each morning and placed into individual Drosophila Assay Monitoring (DAM) chambers (TriKinetics Inc, Waltham, MA). The DAM monitors were placed into a dark incubator. Two hours later, a 500 lux light was turned on by a timer for five minutes. Data was collected with DAMSystem3.0 and DAMFileScan11.0 (TriKinetics Inc). The resulting data was exported to Microsoft Excel and graphed in GraphPad Prism. Three replica experiments were averaged and plotted as Time (min) vs Average activity per 2 min. bin (n = 3). The change in response to light was calculated for each light pulse as [*mean beam breaks for 10 min. post-pulse*] – [*mean beam breaks for 10 min. pre-pulse*].

## Scanning electron microscopy

SEMs of fly eyes were obtained as previously described (*Wolff, 2011*). Eyes were fixed in 2% paraformaldehyde, 2% glutaraldehyde, 0.2% Tween 20, and 0.1 M cacodylate buffer, pH 7.4, for 2 hr. Samples were washed four times with increasing ethanol (25–100%) for 12 hr each followed by a series of hexamethyldisilazane washes (25–100% in ethanol) for one hour each. Flies were air dried for 24 hr, mounted on SEM stubs, and the bodies were coated in fast-drying silver paint. Flies were sputter coated with a gold/pallidum mixture for 60 s and imaged at 900X magnification, with extra high tension set at 3.0 kV on a scanning electron microscope (Sigma SEM; Carl Zeiss, Germany). Ten flies per genotype were mounted and three were imaged (n = 3). Blinding of the samples' identity to the user acquiring the images was performed.

## Transmission electron microscopy

TEMs of retina sections were performed as previously described (*Jenny, 2011*; *Rahman et al., 2012*). Briefly, 550 nm sections were cut and stained with toluidine blue to confirm orientation and section depth. Blocks were subsequently thin-sectioned at 70 nm with a diamond knife (Diatome, Hatfield, PA) on a Leica Ultracut six ultramicrotome (Leica Microsystems, Wetzlar, Germany) and collected onto formvar-coated, glow-discharged copper grids, post-stained with 2% aqueous uranyl acetate and lead citrate. Images were acquired on a Tecnai G2 spirit transmission electron microscope (FEI) equipped with a LaB6 source using a voltage of 120 kV. Blinding of the samples to the technicians performing the processing and the user acquiring the images was performed. Two fly heads per genotype/condition and at least three thin sections per sample were examined (n = 2). Samples were unmasked after the images were processed.

## Immunohistochemistry for BiP and UPR reporters

Fly heads were dissected in HL3 hemolymph-like solution, fixed for four hours in ice-cold 4% paraformaldehyde in filtered PBS, washed overnight in 25% (wt/vol) sucrose in phosphate buffer (pH 7.4), embedded in Optimal Cutting Temperature compound (EMS, Hatfield, PA), frozen in dry ice and sectioned at 20 µm thickness on a cryostat microtome (CM 1950, Leica Microsystems, Wetzlar, Germany). Sections were probed overnight with primary antibodies against Drosophila BiP (1:2000, Gift from Don Ryoo (*Ryoo et al., 2007*), GFP (1:1000, A10262, ThermoFisher Scientific, Waltham, MA) or RFP (1:1000, 600-401-379, Rockland, Limerick, PA). Secondary antibodies were labeled with Alexa488-conjugated Goat anti-Chicken (Molecular Probes, P/N# A-11039), Alexa488-conjugated Goat anti-Guinea Pig (Molecular Probes, P/N# A-11073), or Alexa568-conjugated Goat anti-Rabbit (Molecular Probes, P/N# A-11011). Alexa 647-conjugated phallodin was also added to label Actin for identifying structures. Images were captured with an oil-immersion 63 × NA−1.4 lens on an inverted confocal microscope (LSM710, Carl Zeiss). For each genotype and light rearing conditions, immunohistochemistry experiments were performed in two biological replicas with new sets of flies, using identical acquisition settings. Blinding of the samples to the user acquiring the images was performed when appropriate.

## Quantification of fluorescence staining

Fluorescence images were quantified using ImageJ (NIH) adapting previous methods (*Nandi et al., 2017*). For each antibody, a threshold was determined, removing the lowest 10% of signal in LD control samples (to reduce variation from low level background signals). This same threshold was applied, and a mask was created for every image in a batch of staining. Within a 1 μm optical slice, the retina and lamina regions were selected manually using an Actin stain and assigned as Regions of Interest. The integrated pixel intensity per unit area was measured within this selected area, redirecting to the threshold mask. In each fly, four sequential optical slices were quantified and averaged. For each genotype and treatment, four flies were quantified from two independent biological replicas for a total of eight flies. Data was normalized to the wild-type LD control for each replica. Outliers of greater than three standard deviations were omitted from the analysis.

## Yeast plasmids and strains

Yeast genetic techniques were performed by standard procedures described previously. (*Sherman et al., 1981*). All strains were cultured in either rich (YPD: 1% yeast extract, 2% peptone, and 2% glucose) or complete synthetic minimal (CSM) media lacking appropriate amino acids with 2% glucose. Yeast were grown to log phase, serially diluted, and spotted onto agar plates to assay fitness and temperature sensitivity as previously described (*Tran et al., 2007*).

DNA fragments of KAR2 was generated by PCR amplification of the endogenous KAR2 gene using the primers 5'-GCATCCGCGGATACTCTCGTACCCTGCCGC-3' and 5'-ATGCGAGCTCCGTA TATACTCAGTATAATC-3'. Plasmid pKAR2:LEU2 and pKAR2:URA3 were generated by subcloning genomic DNA fragments containing promoter and coding sequence of *KAR2* into the *SacI* and *SacII* sites of pRS315 and pRS316, respectively. pKAR2T386A:LEU2 was generated by site directed mutagenesis of pKAR2:LEU2 using the primers 5'-GGTTGGTGGTTCTGCTAGAATTCCAAAGGTCCAA-CAATTGTTAGAATCATACTTTGATGG-3' and 5'-ACCTTTGGAATTC TAGCAGAACCACCAACCAAAACGATATCATCAACATCCTTCTTTTCC-3'. pKAR2T538A:LEU2 was generated by site directed mutagenesis of pKAR2:LEU2 using the primers 5'-AGATAAGGGAGC TGGTAAATCCGAATCTATCACCATCACTAACG-3' and 5'-GGATTTACCAGCTCCCTTATCTG TGGCAGACACCTTCAGAATACC-3'.

ACY008 yeast (mat A kar2::KANΔ his3Δ0 leu2Δ0 LYS met15Δ0 ura3Δ0 pKAR2:URA) were obtained by sporulation and dissection of KAR2 heterozygous null yeast (Mata/mat@ KAR2::KAN/KAR2 his3Δ0/his3Δ0 leu2Δ0/leu2Δ0 LYS/lys MET/met15Δ0 ura3Δ0/ura3Δ0) (GE) transformed with pKAR2: URA. Standard plasmid shuffle techniques with 5-FOA(Zymo) were utilized to obtain ACY016 (mat A kar2::KAN his3Δ0 leu2Δ0 LYS met15Δ0 ura3Δ0 pKAR2:LEU2) ACY017(mat A kar2::KAN his3Δ0 leu2Δ0 LYS met15Δ0 ura3Δ0 pKAR2T386A:LEU2), and ACY020(mat A kar2::KAN his3Δ0 leu2Δ0 LYS met15Δ0 ura3Δ0 pKAR2T538A:LEU2)

## Statistics

Statistics were performed using GraphPad Prism 7. Normality of data distribution was determined using D'Agostino's and Pearson's normality test. For the genetic analysis in *Figure 1* and the ERG measurements in *Figure 1—figure supplement 2*, *Figure 1—figure supplement 3*, *Figure 3—figure supplement 2*, *Figure 3—figure supplement 3*, significance was determined using one-way ANOVA, followed by Tukey's multiple comparisons tests. Statistical significance for non-parametric data, including the ERGs with light treatment quantifications in *Figure 2*, was determined by Kruskal-Wallis tests followed by multiple comparisons testing with Dunn's correction. For the image quantification data in *Figure 4*, significance was determined by two-way ANOVA followed by multiple comparisons with Benjamini-Krieger-Yekutieli's False Discovery Rate correction. All tests were two-sided with no experimental matching. RStudio (version 1.1.442, 2018, RStudio, Inc.) was used for Fisher's Exact Tests for the eye interaction screen, with Bonferroni's multiple comparison method to determine significance (*Figure 2—figure supplement 1*). Tests were two-sided. When possible, blinding of sample identities was performed for image acquisition and fluorescence intensity quantification. Sample sizes for ERG assays, EM experiments, fluorescence quantifications and fly genetic analysis were determined based from previous experience (*Rahman et al., 2012*; *Stenesen et al., 2015*; *Nandi et al., 2017*).

## Acknowledgements

We thank Drs. Eric Olson and Joe Takahashi and the members of the Krämer and Orth labs for discussion and technical assistance. We thank the Bloomington Stock Center (NIH P40OD018537) and the Vienna Drosophila Resource Center (VDRC, www.vdrc.at) for flies and the Molecular and Cellular Imaging Facility at the University of Texas Southwestern Medical center for help with electron microscopy (NIH S10 OD020103-01). KO is a Burroughs Welcome Investigator in Pathogenesis of Infectious Disease, a Beckman Young Investigator, and a W W Caruth, Jr., Biomedical Scholar and has an Earl A Forsythe Chair in Biomedical Science.

## Additional information

### Competing interests

Kim Orth: Reviewing editor, *eLife*. The other authors declare that no competing interests exist.

### Funding

| Funder | Grant reference number | Author |
| --- | --- | --- |
| National Institute of General Medical Sciences | R01GM120196 | Helmut Krämer |
| National Eye Institute | RO1EY010199 | Helmut Krämer |
| Howard Hughes Medical Institute | | Kim Orth |
| Welch Foundation | I-1561 | Kim Orth |
| Once Upon A Time Foundation | | Kim Orth |
| National Science Foundation | 1000176311 | Andrew T Moehlman |
| National Institute of General Medical Sciences | RO1GM115188 | Kim Orth |

The funders had no role in study design, data collection and interpretation, or the decision to submit the work for publication.

### Author contributions

Andrew T Moehlman, Conceptualization, Data curation, Validation, Investigation, Methodology, Writing—original draft, Writing—review and editing; Amanda K Casey, Conceptualization, Data curation, Investigation, Writing—review and editing; Kelly Servage, Investigation, Writing—review and editing; Kim Orth, Conceptualization, Supervision, Funding acquisition, Project administration, Writing—review and editing; Helmut Krämer, Conceptualization, Data curation, Supervision, Funding acquisition, Investigation, Project administration, Writing—review and editing

### Author ORCIDs

Andrew T Moehlman http://orcid.org/0000-0002-9233-5515
Kim Orth http://orcid.org/0000-0002-0678-7620
Helmut Krämer http://orcid.org/0000-0002-1167-2676

### Decision letter and Author response

Decision letter https://doi.org/10.7554/eLife.38752.025
Author response https://doi.org/10.7554/eLife.38752.026

## Additional files

### Supplementary files

• Transparent reporting form

DOI: https://doi.org/10.7554/eLife.38752.023

**Data availability**

All data generated or analysed during this study are included in the manuscript and supporting files.

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
