## [Decision Letter]

Thank you for submitting your article "Regulation of the Unfolded Protein Response by BiP AMPylation protects photoreceptors from light-dependent degeneration" for consideration by *eLife*. Your article has been reviewed by three peer reviewers, and the evaluation has been overseen by a Reviewing Editor and K VijayRaghavan as the Senior Editor. The following individual involved in review of your submission has agreed to reveal her identity: Shaeri Mukherjee (Reviewer #2).

The reviewers have discussed the reviews with one another and the Reviewing Editor has drafted this decision to help you prepare a revised submission.

Summary:

The manuscript presents an analysis of the *Drosophila* photosystem in mutants with targeted mutations in Fic, an enzyme that mediates reversible AMPylation of BiP in the ER. The authors report that BiP is a major substrate of in vivo Fic activity and that this activity is required for maintaining the structure and function of photoreceptor neurons in a stress condition (constant light). Genetic interactions between *fic* and BiP point mutants with mutations of UPR components lead the authors to conclude that Fic-mediated BiP AMPYlation regulates the UPR.

Essential revisions:

The reviewers uniformly noted the significance and high quality of the work. A major point of concern was raised by all three reviewers independently regarding the lack of direct evidence for regulation of the UPR by Fic and BiP in the context of photoreceptor degeneration. It is shown that the UPR is activated in the *fic* and *BiP*^T366A^ mutants, but the intervening steps linking these are not elucidated. Given that BiP can mediate ER processes other than UPR, the degeneration and functional defects observed in those mutants could be in parallel instead of consequent to dysregulation of UPR. Nonetheless, I feel that your interpretation that unregulated UPR in the *fic* and *BiP*^T366A^ mutants is a plausible and likely explanation for observed light-dependent degeneration of photoreceptors. Accordingly, with the agreement of the reviewers, before accepting your paper, it will be necessary for you to revise the title and text to indicate that the results do not demonstrate that BiP AMPylation directly regulates the UPR. In addition, it is advisable to include in the Discussion other plausible models to explain the data.

---

## [Author Response]

Essential revisions:The reviewers uniformly noted the significance and high quality of the work. A major point of concern was raised by all three reviewers independently regarding the lack of direct evidence for regulation of the UPR by Fic and BiP in the context of photoreceptor degeneration. It is shown that the UPR is activated in the fic and BiP^T366A^ mutants, but the intervening steps linking these are not elucidated. Given that BiP can mediate ER processes other than UPR, the degeneration and functional defects observed in those mutants could be in parallel instead of consequent to dysregulation of UPR. Nonetheless, I feel that your interpretation that unregulated UPR in the fic and BiP^T366A^ mutants is a plausible and likely explanation for observed light-dependent degeneration of photoreceptors. Accordingly, with the agreement of the reviewers, before accepting your paper, it will be necessary for you to revise the title and text to indicate that the results do not demonstrate that BiP AMPylation directly regulates the UPR. In addition, it is advisable to include in the Discussion other plausible models to explain the data.

We agree that we have not shown and don’t want to imply that the excessive activation of the UPR is the driving factor causing the structural and functional changes we observed in the *fic* and *BiP*^T366A^ mutants in response to prolonged light exposure. Accordingly, we made several changes in the manuscript to clarify this issue:

First, we changed the title to say:

“Adaptation to constant light requires Fic-mediated AMPylation of BiP to protect against reversible photoreceptor degeneration”.

Second, we added a section in the Discussion to stress multiple mechanisms that could contribute to the structural and functional changes we describe:

“It is unknown whether the ratio of AMPylated to unmodified BiP directly regulates activation of the UPR sensors, given the multiple levels of feedback on the system (Gardner et al., 2013). It remains to be determined whether the effects on UPR signaling are the direct cause of the visual system defects or if the phenotypes in *fic* and *BiP*^T366A^ mutants are primarily due to changes in protein folding and secretion. We also cannot rule out the influence of AMPylation on other ER processes that involve BiP, such as ER-associated degradation (Hegde et al., 2006).”

Furthermore, we changed a few specific sentences throughout the manuscript along those lines. Specifically:

At the end of the Introduction, it now says:

“In addition, we identify changes in the regulation of UPR during constant light stress in these mutants, implicating dysregulation of ER homeostasis as a probable cause of the inability to adapt to altered light conditions.”

The subtitle in the Results section introducing our findings regarding the dysregulated UPR, now says: a

“Fic is required for ER homeostasis in the visual system during constant light”.

The end of the Results section, now states:

“Together, these data identify a role for Fic-mediated BiP AMPylation in regulating ER stress during homeostatic responses of the visual system.”

The title of the figure describing the UPR results, now states:

“Figure 5. ER homeostasis is disturbed in *fic* mutants during prolonged light stimulation.”